# School-based interventions to prevent anxiety and depression in children and adolescents in low- and middle-income countries: A systematic review

Sharone Zhameden Dieu Yin[1], Mei Ken Low[2], Masuma Pervin Mishu[3]*

1 Institute of Epidemiology and Health Care, University College London, London, United Kingdom,
2 Institute of Ophthalmology, University College London, London, United Kingdom, 3 Department of Epidemiology and Public Health, Institute of Epidemiology and Health Care, University College London, London, United Kingdom

* masuma.mishu@ucl.ac.uk

## Abstract

### Background

Anxiety and depression are on the rise among children and adolescents globally. Low- and middle-income countries (LMICs) face heightened vulnerability due to limited resources, restricted access to mental health services, socioeconomic disparities and widespread mental health stigma. Schools offer a unique and potentially impactful setting for preventive interventions targeting anxiety and depression in young individuals.

### Aim

We aimed to identify empirical research to explore the effectiveness of school-based interventions designed to prevent anxiety and depression among children and adolescents in LMICs.

### Method

Ovid MEDLINE, Embase, PsycINFO and CENTRAL were systematically searched for articles published between 2018 and July 2023. Randomised controlled trials that evaluated school-based interventions for children and adolescents aged 4–18 years in LMICs were included. Only studies in English language were included. The primary outcomes were anxiety and/or depressive symptoms. Risk-of-bias assessments were performed.

### Results

Out of 3863 articles identified, six studies comprising 1587 students met the inclusion criteria. Among the four studies that examined interventions for the prevention of both anxiety and depression, as well as anxiety alone, only one study showed a reduction in anxiety symptoms. In the case of depression, three out of four studies reported improvements in depressive symptoms. The finding suggests a potential effectiveness of

**Data availability statement:** All relevant data are within the paper and its Supporting Information files.

**Funding:** The author(s) received no specific funding for this work.

**Competing interests:** The authors declare no conflict of interest

preventive interventions against depression, but not anxiety. However, this finding should be interpreted with caution given the limited number of studies identified. All studies were either classified as high risk of bias or having some concerns.

## Conclusion

There is some evidence of the effectiveness of school-based interventions in preventing anxiety and depression among young people in LMICs. Further research is necessary to gain a more comprehensive understanding of this critical issue. Moving forward, it is crucial to enhance and broaden existing school-based prevention programs in these nations, exploring different intervention strategies tailored to their specific contextual factors.

## Introduction

Mental health problems among children and adolescents are major public health concerns globally. According to the World Mental Health Report published by World Health Organization (WHO) in 2022, approximately 8% of young children aged 5–9 years and 14% of adolescents aged 10–19 years are affected by mental disorders globally [1]. The most common mental health disorders are anxiety and depression, accounting for over 40% of cases among adolescents [2]. Since the onset of the coronavirus disease 2019 (COVID-19) pandemic, there has been a doubling in the estimated rates of anxiety and depression symptoms among children and adolescents. A meta-analysis of 29 studies revealed that the global prevalence of clinically elevated anxiety and depressive symptoms in this age group during the first year of the pandemic were 20.5% and 25.2%, respectively [3]. The pandemic has exacerbated existing mental health challenges and introduced new stressors for this demographic. Factors such as isolation, disrupted routines and education, financial strain and increased feelings of uncertainty about the future have significantly contributed to heightened levels of psychological distress [4–6].

According to the most recent systematic analysis of the global burden of disease among young people aged 10–24 years, self-harm, a severe outcome strongly linked to poor mental health, has emerged as one of the leading causes of adolescent mortality. It contributes to 8.2% of all adolescent deaths worldwide and approximately 20% of deaths within the age group of 15–24 years [7]. The majority of mental health disorders begin in adolescence, with half of them starting by the age of 14 and three-quarters by the age of 25 [8,9]. Research indicates that mental health issues experienced during childhood and adolescence can persist or recur over the lifespan [10–12] and have long-term biopsychosocial ramifications [13–15]. Furthermore, childhood mental disorders have a significant economic impact [16–18]. According to the United Nations Children's Fund's (UNICEF) State of the World's Children 2021 report, mental health conditions in children aged 0–19 lead an estimated annual loss of US$340.2 billion, adjusted for purchasing power parity, in human capital [2].

The burden of mental health problems is unevenly distributed, with a higher prevalence observed in low- and middle-income countries (LMICs). It is estimated that over 225 million children and adolescents worldwide are affected by mental disorders and 88% of these cases are found in LMICs [19]. However, there is limited data on mental health, especially for children and adolescents, in these regions. Despite LMICs being home to nearly 90% of the global adolescent population, data coverage is only available for approximately 2% of children and adolescents living in these countries [1,2,20]. Children and adolescents in LMICs face distinct challenges when it comes to addressing mental health issues and providing mental health care

services. These challenges include limited human and financial resources, inadequate mental health infrastructure and vast variations in socio-cultural contexts [21]. Health systems often exhibit inadequate integration of mental health services into the broader healthcare policies or within primary healthcare [22–24]. Additionally, this population faces a higher risk of being exposed to adverse childhood experiences and various risk factors such as physical and emotional abuse, chronic neglect, poverty, homelessness, violence and conflicts [25–30]. Furthermore, a culture of stigma surrounding mental health is widespread in LMICs. Misconceptions, fear and societal attitudes regarding mental illness often led to social exclusion, discrimination and hesitancy to seeking assistance [31–33]. Cultural and religious beliefs about illnesses have driven many individuals to favour non-evidence-based approaches provided by traditional or alternative healers, such as complementary practitioners or spiritual healers, instead of relying on current therapeutic interventions [23,34]. The stigma associated with mental health alongside cultural and religious perceptions have resulted in significant delays in help-seeking, reduced access to health services and poor adherence to treatments.

Given the long-term implications of untreated mental health conditions, there is a critical need for preventive measures and timely interventions. Early prevention strategies are crucial in reducing the prevalence of mental health disorders in children and young people, as well as alleviating the associated burden and disability that comes with these conditions. Primary prevention refers to measures designed to prevent the occurrence of a disease before it develops in a susceptible population or individual [35]. Interventions, aimed at primary prevention, can be classified as universal, selective and indicated. Universal prevention aims to reach the entire population not defined on the basis of risk. On the other hand, selective prevention focuses on individuals who have a higher-than-average risk of developing a mental disorder. Lastly, indicated prevention targets high risk individuals who exhibit early subclinical symptoms of a mental disorder. Risk factors are determined by biological, psychological or social indicators [36]. Policies have prioritised primary prevention of mental disorders in children and young people, with schools playing a central role in implementation [37]. Schools serve as important environments for social and emotional learning [38]. Since education is compulsory in most LMICs [39,40], the majority of children and young people spend a substantial amount of time in schools. Therefore, school-based interventions have the potential to reach a large number of children and adolescents, irrespective of their socio-cultural backgrounds. They can help overcome challenges related to limited resources and poor access to mental health services.

A diverse array of interventions has been integrated into educational settings, falling into categories such as psychological, psychosocial, psychoeducational, psycho-supportive, physical, mindfulness, relaxation and many more [41]. Among these, cognitive-behavioural therapy (CBT) is one of the most widely used psychological interventions, supported by robust evidence demonstrating its efficacy in addressing both anxiety and depression [42]. Other examples of psychological interventions are behavioural therapy (BT), third wave therapies and interpersonal therapies [41].

Several systematic reviews have sought to assess the effectiveness of school-based preventive interventions in addressing anxiety and depression among children and adolescents [41,43–46]. However, most of the interventions have been evaluated in high-income countries (HICs). Considering the unique mental health challenges and disparities in LMICs, it remains uncertain whether these interventions can be effectively applied in such settings. A comprehensive review of preventive interventions, accounting the specific contexts of LMICs, is necessary to inform ongoing policy implementation in these countries.

This systematic review aims to identify and explore the effectiveness of interventions to prevent anxiety and depression in children and adolescents in LMICs.

## Methods

This systematic review was conducted and reported according to the preferred reporting items for systemic reviews and meta-analyses (PRISMA) guidelines [47]. The protocol was registered retrospectively on INPLASY: INPLASY2024100063 on 15th October 2024.

### Eligibility criteria

**Inclusion criteria.** Studies with participants aged between 4 and 18 years at recruitment and who did not have an identifiable physical or mental health condition were included. Studies were eligible if they evaluated school-based interventions aimed at preventing anxiety and depression. These interventions were required to take place in school settings or be integrated into the school curriculum. Various primary prevention approaches, including universal, selective or indicated prevention, were eligible. Randomised controlled trials (RCTs) including both individual and cluster RCTs reporting anxiety and/or depression symptoms as the outcomes were included. All types of outcome assessors, such as participants, parents, teachers and clinicians, were eligible. All types of control group were eligible. These included no intervention (NI), usual curriculum (UC), waitlist (WL) or attention control (AC) groups. Only studies conducted in LMICs, based on the Development Assistance Committee (DAC) list of recipients eligible for official development assistance (ODA) funding [48], were included. Studies published from 2018 were included. The date limit was selected because the search strategy of the last major systematic review [41] on school-based anxiety and depression prevention interventions was conducted until 2018. Studies published in English Language were included.

**Exclusion criteria.** Studies that focused on addressing mental health promotion, awareness or literacy, emotional well-being and positive psychology were not eligible, unless their aim was to prevent anxiety and depression. Interventions intended to address problems potentially leading to a mental health disorder, for example, stress, bullying and substance abuse, were excluded. Similarly, interventions aimed to help children and young people cope with specific events or circumstances such as parental divorce, natural disasters and conflicts, were excluded. Any grey literature and reviews, reports and study protocols were excluded.

### Search strategy and study selection

Ovid MEDLINE, Embase, PsycINFO and CENTRAL were searched on 7th July 2023. The search terms were developed using Caldwell et al.'s search strategy [41] and further expanded to encompass LMICs listed in the DAC list of ODA recipients [48]. The concepts used in the structured search were "children and young people", "school-based", "depression and anxiety", "preventive interventions", "risk factors", "randomised controlled trials" and "low- and middle-income countries". Under these concepts, relevant keywords, synonyms and Medical Subject Headings (MeSH) terms were included. Boolean operators "OR" was used to link search terms within each concept and "AND" was used to combine the different concepts. The search strategy was checked by an information specialist and modified for each database [49].

Titles, abstracts and full texts were screened fully by the primary reviewer, SZDY. A second reviewer, MKL, independently undertook 20% of titles, abstracts and full-texts screening. Discrepancies were jointly reconciled by the reviewers, SZDY and MKL. Any disagreement was resolved by a third reviewer, MPM, when necessary. References were collated using Endnote [50] and transferred to Rayyan [51] for screening.

### Data extraction

A data extraction form was developed to register and code relevant information about the included studies. Data on study details (authors, year of publication, study design, country,

study setting, target condition, and type of prevention intervention), participant characteristics (age range, gender and sample size), interventions (program name, mode of delivery, program format, program content, who delivered the intervention, number and duration of sessions), controls (types and content of control) and outcomes (duration of follow-up, outcome assessors, measurement scale used, pre- and post-intervention results and effectiveness of intervention on symptoms of anxiety and/or depression) were extracted.

Data were extracted by the primary reviewer, SZDY, and checked by a second reviewer, MPM.

### Risk of bias and quality assessment

The Cochrane Risk of Bias tool version 2 (RoB 2 tool) [52] was used to assess the risk of bias of all included studies. The RoB 2 for cluster-randomised trials was employed when the study design involved a cluster RCT. Studies were rated as 'low', 'some concerns' and 'high' in risk of bias. The process of assessment was performed by the primary assessor, SZDY, and were checked by a second assessor, MPM. Discrepancies were resolved through discussion.

The quality of evidence was assessed using the Grading of Recommendations Assessment, Development and Evaluation (GRADE) approach [53]. Studies were classified as 'high', 'moderate', 'low' and 'very low' in quality of evidence. The quality assessment of included studies was performed independently by two assessors, SZDY, and MKL. Discrepancies were resolved through discussion.

### Data Synthesis

Due to the heterogeneity of the included studies, it was not possible to conduct a meta-analysis. A narrative synthesis of anxiety and/or depression outcomes of all included studies was undertaken. For reporting of anxiety and/or depression outcomes, we grouped the studies as they reported the outcome as: anxiety and depression, anxiety only and depression only.

## Results

### Study Selection

A total of 3863 records were identified through electronic database searching, of which 1102 duplicates and one retracted paper were removed. Following a 100% agreement rate in the 20% double-screened abstracts, 2760 records underwent title and abstract screening, resulting in the exclusion of 2711 records. 49 full-text articles were assessed for eligibility. Finally, six studies [54–59] were included in the review, encompassing data from 1587 participants. Fig 1 shows the PRISMA flow diagram of the selection process.

### Study characteristics

Table 1 shows the characteristics of the included studies. The included studies were published between 2018 and 2021. Sample sizes ranged from 43 to 580 individuals with the age range between 4–18 years. Four studies were cluster-RCTs and two were individually randomised, parallel trials. All studies were conducted in middle-income countries (MICs); four [54–56,58] were conducted in upper and two [57,59] were conducted in lower MICs. The upper MICs were Brazil [58], China [55], Lebanon [56] and Malaysia [54] while the lower MICs were India [59] and Kenya [57]. Five studies were classified as universal prevention [54–58], and one as indicated [59]. Among the six studies that were included, one study took place in a pre-school setting [58], another in a primary school setting [54] and the majority were conducted in secondary schools (n=4) [55–57,59]. Studies evaluated interventions that target anxiety (n=2) [54,58], depression (n=2) [55,59] and both anxiety and depression (n=2) [56,57].

**Fig 1. PRISMA flow diagram.**

The preventive interventions were based on CBT (n=4) [54,56,58,59] and psychoeducation (n=2) [55,57] programs. The program sessions varied in number, spanning from 1 to 18, and the duration of each session ranged between 40–120 minutes. Regarding control conditions, three studies utilised the wait-list control condition [54,56,58], two used the attention control condition [57,59] and one used the usual curriculum [55] condition. Studies were delivered by teachers (n=1) [55], mental health professionals (n=2) [56,58] and researchers (n=3) [54,57,59]. Mental health professionals were psychologists, psychiatry trainees, post-doctoral research fellows and master's students in psychology. Five studies [54–56,58,59] were delivered face-to-face whereas one study [57] utilised digital means as mode of delivery. All were delivered through group settings and included a mixture of both genders.

**Table 1. Study characteristics.**

| Author and Year | Study Design | Target Condition | Country | Study Setting | Prevention Type | Age Range | Sample Size | Program Name | Program Content | Mode and Format of Delivery | Delivered By | Number, Duration of Sessions | Control Types and Content |
|---|---|---|---|---|---|---|---|---|---|---|---|---|---|
| Ab Ghaffar 2019 | Cluster RCT | A | Malaysia | Primary | U | 10-11 | 461 | IMB-based prevention program | CBT | F2F, Group | Research Assistants | 4, 60 mins | WL |
| Desan 2021 | Cluster RCT (Pilot) | D | China | Secondary | U | NR (US 10th Grade) | 580 | Curriculum based on positive psychology | Psycho-education | F2F, Group | Teachers | 18, 40 mins | UC – traditional psychology curriculum |
| Maalouf 2020 | Cluster RCT | A and D | Lebanon | Secondary | U | 11-13 | 280 | My FRIENDS Youth program | CBT | F2F, Group | Mental Health Professionals or Trainees | 10, 45–50 mins | WL |
| Osborn 2020 | Individual RCT | A and D | Kenya | Secondary | U | 13-18 | 103 | Shamiri-Digital | Psycho-education | MM, Group | Student Researchers | 1, 90 mins | AC – active, digital study-skills |
| Rivero 2020 | Individual RCT (Pilot) | A | Brazil | Pre-School | U | 4-6 | 43 | FunFRIENDS program | CBT | F2F, Group | Psychologists | 14, 90–120 mins | WL |
| Singhal 2018 | Cluster RCT | D | India | Secondary | I | 13-18 | 120 | Coping Skills program | CBT | F2F, Group | Researchers | 8, NR | AC – single interactive psycho-education |

RCT – Randomised Controlled Trial, A – Anxiety, D – Depression, A and D – Anxiety and Depression, U – Universal, I – Indicated, IMB – Information-Motivation-Behaviour, CBT – Cognitive Behavioural Therapy, F2F – Face-to-face, MM – Multimedia, UC – Usual Curriculum, WL – Waiting List, AC – Attention Control. NR-Not reported

Table 2 displays the outcome characteristics of the included studies. Five studies evaluated the effects of the prevention programs on anxiety and/or depression by utilising measures based on self-reported symptoms. One study, however, employed measures reported by parents. Four studies [54–56,59] reported an end of intervention outcome on anxiety and/or depression symptoms. Additionally, two [57,58] presented follow-up data either before or at the one-month timepoint, while three studies [54,58,59] presented follow-up data at the three-month timepoint. Most programs (n=5) did not report any parental involvements. In the study conducted by Rivero et. al. [58], parents were provided guidance for fostering resilience within the family setting, building healthy habits, parental strategies and enhancing social-emotional skills. Additionally, they received a weekly report outlining the skills cultivated during each FunFRIENDS session and instructions for activities to be performed at home.

Due to the heterogeneity in the studies and utilisation of diverse outcome measures, a meta-analysis for combining the results of the included studies was not feasible. We conducted a narrative synthesis of the results, and the effectiveness of the interventions across the studies was reported individually. The effectiveness of the interventions reported in the included studies are presented based on the specific target conditions: the combination of anxiety and depression, anxiety alone and depression alone, as outlined in Table 2. Among these, four studies [54,56,57,59] reported statistically significant improvements in either anxiety or depression symptoms within the intervention group when compared to the comparator group.

## Anxiety and depression

Two studies evaluated interventions that targeted both anxiety and depression in children and young people. In the cluster RCT conducted by Maalouf and colleagues [56], a total

**Table 2. Outcome measures and results of included studies.**

| Study | Outcome Measures (Cut-off Scores) | Intervention Group (Mean, SD, n) | | | Control Group (Mean, SD, n) | | | Overall Results |
|---|---|---|---|---|---|---|---|---|
| | | Pre-intervention | Post-intervention (Intervention Endpoint) | Follow-up | Pre-intervention | Post-intervention (Intervention Endpoint) | Follow-up | |
| **Studies that evaluated interventions on both anxiety and depression** | | | | | | | | |
| Maalouf 2020 | Anxiety: Scale for Childhood Anxiety and Related Disorders (SCARED) (No Cut-off identified)[*1] | 31.24 (-), n=144 | 23.16 (-), n=102 | – | 26.24 (-), n=133 | 19.66 (-), n=126 | – | No significant time*group effect was found for the total SCARED score (p=0.709). |
| | Depression: Mood and Feelings Questionnaire (MFQ) (No Cut-off identified)[*2] | 18.46 (-), n=144 | 10.66 (-), n=102 | – | 14.13 (-), n=133 | 9.36 (-), n=126 | – | Significant time × group effect for the MFQ for depressive symptoms (p= 0.039) scores, indicating that being in the intervention group was associated with a significant decrease in emotional and depressive symptoms compared with the control group over time. |
| Osborn 2020 | Anxiety: GAD-7 (Cut-off: ≥10) | 8.98 (5.12), n=50 | – | 2 weeks 7.92 (4.48), n=50 | 8.74 (5.30), n=53 | – | 2 weeks 9.00 (4.45), n=53 | Although adolescents in the intervention group experienced a decline in anxiety symptoms from baseline to 2-week follow-up compared with the control group youths, this decline was non-significant (p=0.280, d=0.29, 95% CI [-0.20, 0.79]). |
| | Depression: PHQ-8 (Cut-off: ≥10)[3] | 10.60 (5.37), n=50 | – | 2 weeks 8.35 (4.69), n=50 | 9.68 (4.75), n=53 | – | 2 weeks 10.00 (4.65), n=53 | The significant Time*Condition interaction indicated that adolescents in the Shamiri-Digital intervention experienced larger declines in depressive symptoms from baseline to 2-week follow-up than control-group youths (p=0.028, d=0.50, 95% CI [0.00, 1.60]). |
| **Studies that evaluated interventions on anxiety** | | | | | | | | |
| Ab Ghaffar 2019 | Anxiety: RCAD-25 (Cut-off: T score ≥65) | 14.14 (6.19), n=193 | 14.10 (6.42), n=- | 3 months 12.95 (7.09), n=172 | 15.06 (6.95), n=268 | 15.14 (6.33), n=- | 3 months 13.87 (7.18), n=241 | The intervention was effective in reducing anxiety for the whole sample (F(4,1097)=5.86, p=0.001, d=0.103). The findings showed that the effect size of our school-based anxiety prevention program at three months post-intervention was small, with Cohen's d=0.10. |
| Rivero 2020 | Anxiety: PAS* (Cut-off: T score ≥60) | ITT1  41.04 (15.65), n=21 | – | 1 month 36.61 (15.37), n=19 | 36.86 (13.76), n=22 | – | 1 month 33.27 (11.73), n=20 | A reduction in overall scores in both groups was observed for the PAS instrument, but without statistical significance (F=0.14, p=0.86, effect=0.00). |
| | | | | 3 months 31.95 (16.32), n=16 | | | 3 months 29.77 (10.13), n=15 | |
| | | ITT2 | – | 1 month 36.46 (15.25), n=19 | | – | 1 month 33.57 (10.32), n=20 | Non-statistical significance (F=0.70, p=0.47, effect=0.01). |
| | | | | 3 months 29.66 (11.37), n=16 | | | 3 months 29.86 (7.60), n=15 | |

*(Continued)*

**Table 2.** (Continued)

| Study | Outcome Measures (Cut-off Scores) | Intervention Group (Mean, SD, n) | | | Control Group (Mean, SD, n) | | | Overall Results |
|---|---|---|---|---|---|---|---|---|
| | | Pre-intervention | Post-intervention (Intervention Endpoint) | Follow-up | Pre-intervention | Post-intervention (Intervention Endpoint) | Follow-up | |
| **Studies that evaluated interventions on depression** | | | | | | | | |
| Desan 2021 | Depression: Centre for Epidemiological Studies Depression Scale (CES-D) (No Cut-off identified)*,4 | 8.15 (4.66), n=288 | 9.22 (4.96), n=252 | – | 8.38 (4.71), n=292 | 10.11 (4.82), n=263 | – | There was a negative intervention effect on CES-D (not statistically significant). Since this is a measure of negative affect, this represents improved (less depressed) scores associated with the intervention (Mean+/-SE=-0.725+/-0.436, p>0.05, 95% CI [-1.582, 0.1331]). |
| Singhal 2018 | Depression: CDI (Cut offs: >14 and <24) | 22.00 (4.30), n=65 | 10.30 (3.20), n=51 | **3 months** 5.10 (2.30), n=51 | 21.80 (3.50), n=55 | 19.90 (3.10), n=49 | **3 months** 22.20 (3.60), n=49 | The intervention group evidenced clinically significant reductions in depressive symptoms at both post-intervention and follow-up. | F(1,90)=234.2, p<0.001 |
| | Depression: Centre for epidemiological Studies-Depression Scale for Children (CES-DC)5 (No Cut-off identified)* | 29.4 (6.40), n=65 | 15.90 (4.90), n=51 | 9.40 (3.30), n=51 | 29.50 (5.30), n=55 | 28.40 (5.30), n=49 | 29.40 (4.60), n=49 | | F(1,90)=132.5, p<0.001 |

- –Parent-reported, SCARED – Scale for Childhood Anxiety and Related Disorders, MFQ – Mood and Feelings Questionnaire, GAD-7 – Generalized Anxiety Disorder Screener–7, PHQ-8 – Patient Health Questionnaire–8, RCAD-25 – Short version of the Revised Child Anxiety and Depression Scale, PAS – Preschool Anxiety Scale, CES-D – Centre for Epidemiological Studies Depression Scale, CDI – Children's Depression Inventory, CES-DC – Centre for Epidemiological Studies-Depression Scale for Children, ITT1 – Intention-to-treat analysis 1, ITT2 – Intention-to-treat analysis

*No identified cut off scores within the included studies.

1–5 highlights commonly used cut off scores for these outcome measures: 1. A total score of ≥ 25 may indicate the presence of an anxiety disorder. Scores higher than 30 are more specific

2Higher scores on the MFQ suggest more severe depressive symptoms. Scores on the long version range from 0 to 66. Scoring 27 or higher on the long version may indicate the presence of depression in the respondent.

3Higher scores on the MFQ suggest more severe depressive symptoms. Scores on the long version range from 0 to 66. Scoring 27 or higher on the long version may indicate the presence of depression in the respondent.

4Cut-off score for depressive symptoms was ≥10 for the 10-item version. 5. Scores over 15 can be indicative of significant levels of depressive symptoms.

of ten schools comprising 280 students aged 11–13 years were randomly assigned to either the My FRIENDS Youth program, a CBT-based intervention, or a wait-list control group. Pre-intervention and post-intervention anxiety scores were measured using the Scale for Childhood Anxiety and Related Disorders (SCARED). A reduction in the SCARED scores was observed for both the FRIENDS program (pre-intervention: Mean=31.24, SD=Not Reported (NR), post-intervention: Mean=23.16, SD=NR) and the wait-list control (pre-intervention: Mean=26.24, SD=NR, post-intervention: Mean=19.66, SD=NR) groups. However, this reduction was not statistically significant. On the other hand, depression scores were measured using the Mood and Feelings Questionnaire (MFQ) at the same time points. Similarly, the MFQ scores reduced for both the intervention and control groups. Unlike the SCARED scores, there was a statistically significant reduction in the MFQ scores in the My FRIENDS Youth program group (pre-intervention: Mean=18.46,

SD=NR, post-intervention: Mean=10.66, SD=NR) as compared to the wait-list control group (pre-intervention: Mean=14.13, SD=NR, post-intervention: Mean=9.36, SD=NR). My FRIENDS Youth program was effective in improving the depressive symptoms in the participants of the study, but it did not demonstrate the same effectiveness in alleviating anxiety symptoms.

In the study by Osborn and colleagues [57], a group of 103 high school students aged 13–18 years were randomised to either the Shamiri-Digital, a single session of digital |psychoeducation-based program, or an active digital study-skills control intervention. Anxiety and depression scores were evaluated using the Generalised Anxiety Disorder Screener-7 (GAD-7] and the Patient Health Questionnaire-8 (PHQ-8) respectively, both before the intervention and at a two-week follow-up. For anxiety outcomes, participants assigned to the Shamiri-Digital intervention experienced a reduction in GAD-7 scores (pre-intervention: Mean=8.98, SD=5.12, 2-weeks post-intervention: Mean=7.92, SD=4.48). On contrast, the study-skills control group showed an increase in GAD-7 scores (pre-intervention: Mean=8.74, SD=5.30, 2-weeks post-intervention: Mean=9.00, SD=4.45). In terms of depression outcomes, the PHQ-8 scores decreased in the Shamiri-Digital group (pre-intervention: Mean=10.60, SD=5.37, 2-weeks post-intervention: Mean=8.35, SD=4.69) while an increase was observed in the study-skills control group (pre-intervention: Mean=9.68, SD=4.75, 2-weeks post-intervention: Mean=10.00, SD=4.65). Comparing the Shamiri-Digital intervention to the study-skills control, a significant reduction in depressive symptoms among adolescents from baseline to the two-week follow-up was evident (p=0.028, 95% CI [0.00, 1.60], d=0.50), whereas no statistically significant effect was seen on anxiety symptoms (p=0.280, 95% CI [-0.20, 0.79], d=0.29).

In summary, both studies demonstrated that school-based interventions led to a statistically significant reduction in depressive symptoms, while no significant effect was observed on anxiety symptoms.

## Anxiety

Among the six studies that were included, two RCTs assessed interventions solely aimed at addressing anxiety. Ab Ghaffar et. al. [54] conducted a cluster RCT in which they randomised 12 primary schools, involving a total of 461 students, into two groups: one receiving a school-based anxiety prevention program, a CBT-based intervention, and the other consisted of a waitlist control group. Anxiety scores were assessed using the short version of the Revised Child Anxiety and Depression Scale (RCAD-25) at three time points: pre-intervention, post-intervention and at a 3-month follow-up. The anxiety scores demonstrated a decrease across all three time points in the intervention group (pre-intervention: Mean=14.14, SD=6.19, post-intervention: Mean=14.10, SD=6.42, 3-months follow-up: Mean=12.95, SD=7.09). Conversely, the control group indicated a minor escalation in anxiety scores from baseline to post-intervention, followed by a reduction at the 3-month follow-up (pre-intervention: Mean=15.06, SD=6.95, post-intervention: Mean=15.14, SD=6.33, 3-months follow-up: Mean=13.87, SD=7.18). The authors concluded that the school-based anxiety prevention program was effective in reducing anxiety, although the effect of the intervention was small (F(4,1097)=5.86, p=0.001, Cohen's d=0.10).

Rivero et. al. [58] conducted a pilot RCT that involved 43 preschoolers aged 4–6 years. These participants were randomly assigned to either a CBT-based intervention called the FunFRIENDS program or a wait-list control group. Anxiety scores were collected from parents using the Preschool Anxiety Scale (PAS) at three time points: pre-intervention, at 1-month follow-up and 3-months follow-up. Through intention-to-treat analyses, with one using a last-observation-carried-forward (LOCF) method (ITT1) and the other employing

mean substitution (ITT2) for handling missing data, reductions were observed in both the intervention group (ITT1; pre-intervention: Mean=41.04, SD=15.65, 1-month follow-up: Mean=36.61, SD=15.37, 3-months follow-up: Mean=31.95, SD=16.32 and ITT2; pre-intervention: Mean=41.04, SD=15.65, 1-month follow-up: Mean=36.46, SD=15.25, 3-months post-intervention: Mean=29.66, SD=11.37) and the control group (ITT1; pre-intervention: Mean=36.86, SD=13.76, 1-month follow-up: Mean=33.27, SD=11.73, 3-months follow-up: Mean=29.77, SD=10.13 and ITT2; pre-intervention: Mean=36.86, SD=13.76, 1-month follow-up: Mean=33.57, SD=10.32, 3-months post-intervention: Mean=29.86, SD=7.60) across all three time points. However, the observed differences across these time points were not statistically significant (ITT1; F=0.14, p=0.86, effect=0.00 and ITT2; F=0.70, p=0.47, effect=0.01).

In summary, both studies observed a reduction in anxiety symptoms. However, in contrast to the small but statistically significant reduction noted in the study by Ab Ghaffar et. al. [54], the results found by Rivero et al. [58] were not statistically significant.

## Depression

Among the six included studies, two RCTs evaluated interventions exclusively focused on addressing depression. In Desan and colleagues' study [55], they conducted a pilot cluster RCT across two distinct cohorts, encompassing a total of 580 students. In the first cohort, ten classrooms were randomly assigned to either a curriculum centred around psychoeducation on positive psychology or a control group with a traditional psychology curriculum. The second cohort comprised eight classrooms, again randomised into intervention and control groups. Depression scores were measured using the Centre for Epidemiological Studies Depression Scale (CES-D) at pre-intervention and post-intervention. Both the intervention (pre-intervention: Mean=8.15, SD=4.66, post-intervention: Mean=9.22, SD=4.96) and control (pre-intervention: Mean=8.38, SD=4.71, post-intervention: Mean=10.11, SD=4.82) groups demonstrated an increase in CES-D scores. The authors reported a non-statistically significant negative intervention effect, indicating an improvement in depressive symptoms among the intervention group (Intervention effect: Mean=-0.73, SE=0.44, p>0.05, 95% CI [-1.58, 0.13]).

Another study by Singhal et. al. [59] examined a CBT-based indicated prevention intervention known as the Coping Skills program, compared to an attention control group consisting of a single interactive psychoeducation session. This study involved two schools with a total of 120 students aged 13–18 years, who were experiencing subclinical depression. They were randomly assigned to either the intervention or control group. The Children's Depression Inventory (CDI) and Centre for Epidemiological Studies-Depression Scale for Children (CES-DC) were administered at pre-intervention, post-intervention and 3-months follow-up. The intervention group reported a decline in both CDI (pre-intervention: Mean=22.00, SD=4.30, post-intervention: Mean=10.30, SD=3.20, 3-months follow-up: Mean=5.10, SD=2.30) and CES-DC (pre-intervention: Mean=29.4, SD=6.40, post-intervention: Mean=15.90, SD=4.90, 3-months follow-up: Mean=9.40, SD=3.30) scores. In contrast, the control group's CDI (pre-intervention: Mean=21.80, SD=3.50, post-intervention: Mean=19.90, SD=3.10, 3-months follow-up: Mean=22.20, SD=3.60) and CES-D (pre-intervention: Mean=29.50, SD=5.30, post-intervention: Mean=28.40, SD=5.30, 3-months follow-up: Mean=29.40, SD=4.60) scores remained relatively consistent. The authors reported a statistically significant reduction in depressive symptoms across all time points, as indicated by both CDI (F(1,90)=234.2, p<0.001) and CES-DC (F(1,90)=132.5, p<0.001) scores.

Both studies found that school-based interventions led to an improvement in depressive symptoms. However, only results found by Singhal et. al. [59] reached statistical significance.

## Risk of bias assessments

The results from the risk of bias assessment are illustrated in Figs 2–4 Four studies were rated as high risk of bias and two studies had some concerns. The main reasons for studies being classified as high risk of bias were due to lack of blindness in the assessment of outcomes, lack of evidence reported on missing data, significant deviations from the intended interventions and problems with the randomisation process. The main reasons for studies being classified as some concerns were due to lack of pre-specified analysis plans, lack of analysis to correct for missing data, possible deviations from intended intervention due to lack of blindness in people delivering interventions, studies not reporting any procedures to guarantee that outcome assessors were blinded to group assignment as well as allocation concealment. Please see Appendix 2 for the full 'Risk of Bias' assessments for each study, including the evidence used to justify ratings.

## Quality of evidence

The main outcomes assessed were anxiety and depression (Table 3). Overall, the quality of evidence across all outcome domains was deemed very low. Several quality concerns arose from several factors, including risk of bias, indirectness and imprecision. Generally, the risk of bias of the included studies stemmed from a lack of allocation concealment, inadequate blinding

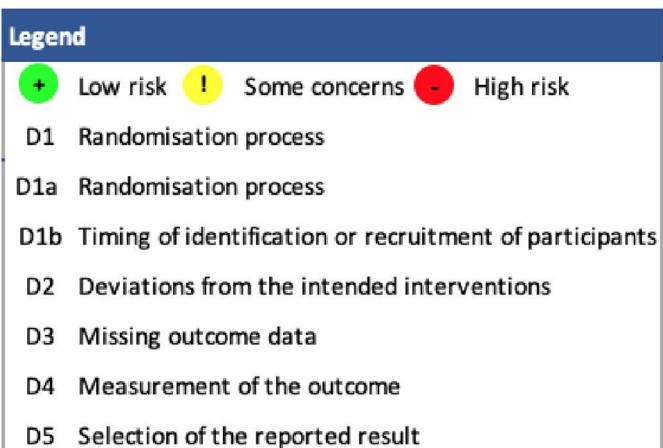

Figure 2: Results of 'Risk of Bias' assessments, categorised based on study design. The RoB 2 Excel tool (Beta Version 9) was used for the generation of part of this figure.

**Fig 2. Results of 'Risk of Bias' assessments, categorised based on study design.** The RoB 2 Excel tool (Beta Version 9) was used for the generation of part of this figure.

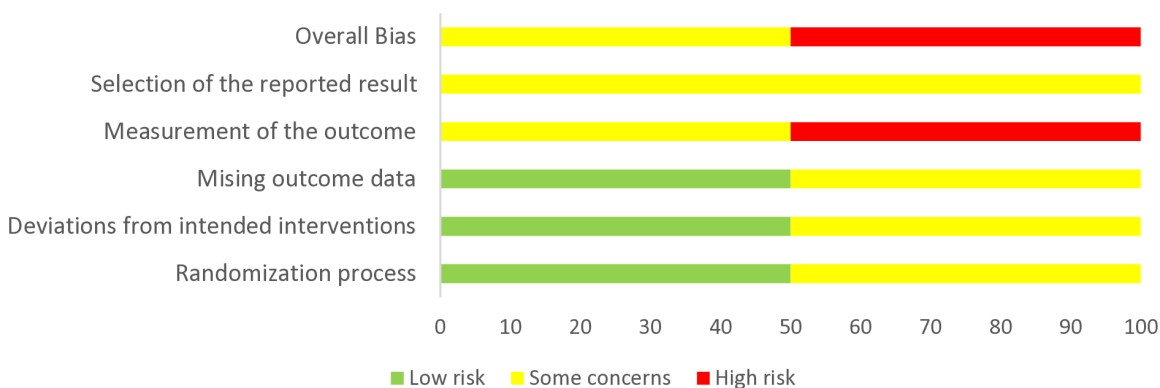

**Fig 3. Judgements for each risk of bias items presented as percentages across all individually randomised parallel-group trials.** Figure generated using RoB 2 Excel tool (Beta Version 9).

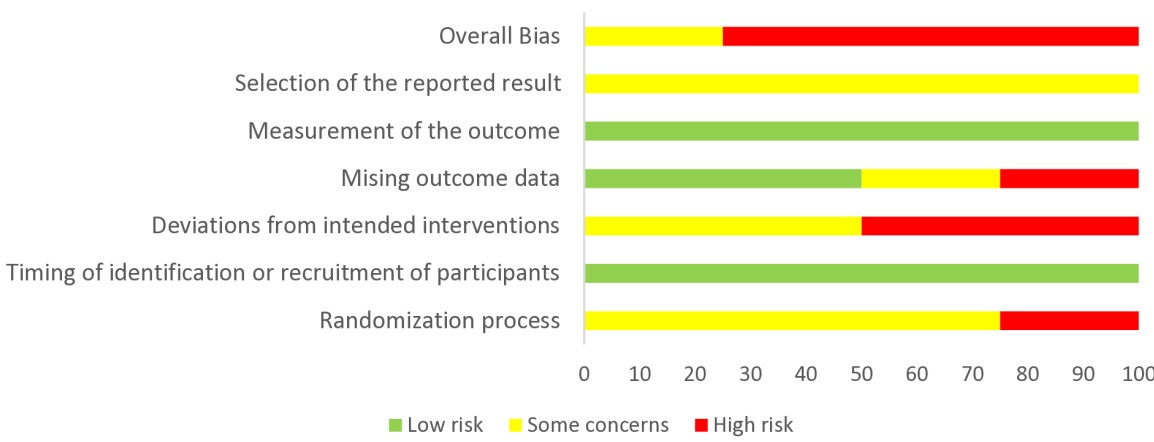

**Fig 4. Judgements for each risk of bias items presented as percentages across all cluster randomised trials.** Figure generated using RoB 2 Excel tool (Beta Version 9).

and incomplete accounting of patients and outcome events. The indirectness of evidence arises from variations in interventions, making their applicability to LMICs uncertain. While the optimal information size was not calculated for the included studies, most had small sample sizes and there was limited information around the precision of results. A funnel plot was not generated due to significant heterogeneity in outcome measures across studies, precluding evaluation of publication bias in the quality assessment.

## Discussion

Overall, six relevant studies of school-based interventions that prevent anxiety and depression among children and adolescents in LMICs were identified. The findings suggest the potential effectiveness of preventive interventions against depression, while the same cannot be concluded for anxiety. In the subset of studies (n=4) that evaluated interventions addressing

**Table 3. Quality assessment of the evidence using the GRADE approach.**

| Outcome | Number of studies | Study design | Risk of bias | Inconsistency | Indirectness | Imprecision | Other factors | Overall |
|---|---|---|---|---|---|---|---|---|
| **Comparison: CBT vs Waitlist** | | | | | | | | |
| Anxiety [54,56,58] | 3 | Cluster and Individual RCTs | Very serious | Serious | Serious [a] | Serious [c] | None | Very low ⊕○○○ |
| Depression [56] | 1 | Cluster RCT | Very serious | Not serious | Serious [a] | Serious [c] | None | Very low ⊕○○○ |
| **Comparison: CBT vs Usual Curriculum** | | | | | | | | |
| Anxiety | 0 | – | – | – | – | – | – | – |
| Depression | 0 | – | – | – | – | – | – | – |
| **Comparison: CBT vs Attention Control** | | | | | | | | |
| Anxiety | 0 | – | – | – | – | – | – | – |
| Depression [59] | 1 | Cluster RCT | Very serious | Not serious | Serious [a] | Serious [c] | None | Very low ⊕○○○ |
| **Comparison: Psychoeducation vs Waitlist** | | | | | | | | |
| Anxiety | 0 | – | – | – | – | – | – | – |
| Depression | 0 | – | – | – | – | – | – | – |
| **Comparison: Psychoeducation vs Usual Curriculum** | | | | | | | | |
| Anxiety | 0 | – | – | – | – | – | – | – |
| Depression [55] | 1 | Cluster RCT | Very serious | Not serious | Serious [a] | Serious [c] | None | Very low ⊕○○○ |
| **Comparison: Psychoeducation vs Attention Control** | | | | | | | | |
| Anxiety [57] | 1 | Individual RCT | Not serious | Not serious | Very serious [b] | Serious [c] | None | Very low ⊕○○○ |
| Depression [57] | 1 | Individual RCT | Not serious | Not serious | Very serious [b] | Serious [c] | None | Very low ⊕○○○ |

– – Not reported, CBT – Cognitive Behavioural Therapy, RCT – Randomised Controlled Trial.

[a].Some variations in interventions adapted to fit specific populations, potentially limiting generalisability to all LMICs

[b].Vast variations in intervention, limiting generalisability to all LMICs

[c].Lack of information regarding adequacy of sample size to demonstrate a precise effect estimate

both anxiety and depression, as well as interventions targeting anxiety alone, only one study [54] demonstrated a statistically significant reduction in anxiety symptoms due to the intervention. In the subset of studies (n=4) that examined interventions targeting both anxiety and depression, as well as interventions exclusively addressing depression, three studies [56,57,59] recorded statistically significant improvements in depressive symptoms. Nevertheless, this finding should be interpreted with caution given the limited number of studies identified. The current review lacked sufficient data to confidently draw conclusions about the overall effectiveness of the identified interventions.

To date, several systematic reviews and meta-analyses [41,43–46] have explored preventative interventions targeting anxiety and depression in children and adolescents within school settings. However, no systematic review that specifically addresses school-based preventative interventions in LMICs has been identified. The overall conclusions drawn from the previous reviews were varied, with most of them indicating a modest beneficial effect in preventing depression and/or anxiety. A meta-analysis performed by Johnstone et. al. [44], involving 14 RCTs with 5970 children under the age of 13, revealed that programs aimed at both anxiety and depression were effective in preventing depressive symptoms, but no significant effect was observed on anxiety symptoms. This aligns with the evidence drawn from the current review, which demonstrates the potential impact of school-based interventions on depressive but not anxiety symptoms.

On the other hand, Corrieri et. al.'s review identified 28 RCTs and found that school-based interventions had a small effect on both depression and anxiety [45]. Similarly, Werner-Seidler et. al. [43] concluded from their systematic review and meta-analysis, involving 81 RCTs with a total of 31 794 participants, that school-based prevention programs showed a small beneficial effect on both depressive and anxiety symptoms. An updated review by Werner-Seidler et. al. in 2021, consisting of 118 RCTs and 45,924 participants, again confirmed its previous findings [46]. In contrast, Caldwell et. al.'s review, encompassing 137 randomised trials with 56 620 participants aged 4–18, reported insufficient evidence to conclude the effectiveness of school-based interventions in preventing anxiety and depression [41]. Among the included studies, only 11 were conducted in LMICs. This finding resonates with the evidence pool identified in this review, where limited data from LMICs is noted. These conflicting conclusions can be attributed to the substantial heterogeneity in the included studies, the variation of methodologies employed and the differences in geographical focus.

Most of the studies included in previous reviews were predominantly centered on HICs, limiting their applicability to LMICs. This is highlighted in a recent overview on depressive disorders prevention in LMICs by Cuijpers et. al. [60], who cautioned against disseminating prevention strategies in LMICs as there is limited studies that have explored its effects in these nations. In Cuijpers' study [60], only 6% of the studies were conducted in LMICs and none of the 50 included RCTs took place in a low-income country (LIC), with only three conducted in MICs. This aligns with our findings, where all studies were based in MICs. LMICs constitute a heterogeneous group of countries with unique contextual factors and implementation challenges that must be considered. Factors such as resource constraints, cultural diversity and mental health stigma in LMICs can significantly influence the design and delivery of interventions, potentially leading to different outcomes from those observed in HICs.

Although there is potential to translate and adapt HICs school-based programs to LMICs, there are several factors that must be considered. Firstly, LMICs may potentially benefit from programs that are cost-effective and less resource-intensive, such as group-based and digital interventions, as suggested by Bradshaw et al. [61]. These programs take into account the limited resources and accessibility needs in LMICs. This has also been reflected in the study by Osborn et. al. [57], which recognised the cost-effectiveness of the digital Shamiri intervention. This contrasts with HICs, where more abundant resources allow for greater flexibility between individual and group interventions.

Another factor of consideration is the method of delivery of the intervention program. The choice to have trained mental health professionals or teachers deliver the school-based intervention, as well as the decision to involve parents, often depends on the availability of resources, cultural values and societal stigma [62]. Heim et. al. [63] demonstrated that cultural values influence the prevalence of mental health disorders. Hierarchy, the legitimisation of inequality, has been depicted to be negatively correlated with mental health disorders, which has been proposed to be due to the stigma of mental health disease found in countries such as China, Turkey and Nigeria. Culturally sensitive school interventions with parental involvement may be especially beneficial in LMICs like China, where family support is a protective factor against mental health disorders [64]. However, this should be balance with the associated cost and resources required [62]. Conversely, in HICs like the United States, public stigma surrounding mental health has declined in recent years [63,64], with egalitarianism potentially contributing to this shift [63]. These differences underscore the importance of tailoring interventions specifically to the social and cultural contexts of both LMICs and HICs.

## Strengths and limitations

One limitation lies in the high heterogeneity observed among the included studies, encompassing differences in their settings, the types of intervention evaluated and how these interventions were delivered. Furthermore, the utilisation of a wide range of measurement tools across studies posed challenges in directly comparing the outcomes, impeding the synthesis of a cohesive overview. Our review highlights that the heterogeneity of outcome measures has posed a major challenge for comparing results and determining the effectiveness of interventions, aligning with previous literature [62,64,65]. Standardising mental health outcome measures would facilitate comparisons across different approaches and programs in future studies [66].

The measurements of anxiety and depression symptoms in all included RCTs relied on subjective reports from participants and parents, as opposed to more reliable clinician-rated outcomes. However, the feasibility of clinicians conducting diagnostic evaluations for all participants in school-based research, particularly given the limited availability of specialised professionals in LMICs, could be challenging. Participant- and parent-reported outcomes are susceptible to reporting and observer bias.

Another limitation stems from the relatively short follow-up periods employed across the studies, with the longest follow-up extending only up to three months after the intervention. The short duration is not sufficient to allow for an assessment of whether the interventions would be effective in preventing the emergence of anxiety and depression.

All studies were classified as having either some concerns or a high risk of bias. This is in part due to the nature of the interventions, making it challenging to implement the double- or triple-blinding conditions typically attributed with high-quality clinical trials. Nevertheless, the risk of bias assessments also revealed other deficiencies, including a lack of information regarding allocation concealment and missing data, as well as insufficient data analysis to address the missing data. Furthermore, none of the included studies had a study protocol that outlined a pre-specified analysis plan to counter the risk of selective reporting. In addition, the GRADE assessment revealed that the quality of evidence for anxiety and depression outcomes were classified as very low. This indicates uncertainty about whether the effects observed in the studies accurately reflect the true impact.

The review's comprehensive search strategy, which encompassed a diverse set of inclusion criteria coupled with the incorporation of risk factors, contributed to a thorough exploration of the topic. While the search strategy was comprehensive, there were limitations that emerged from the complexity and low specificity of the search. One significant limitation was restricting the search to only English language studies. Since many of the LMICs and local research may not be in English. The implementation of English language restriction makes this review susceptible to language bias as relevant studies conducted in LMICs might be published in different languages and would have been excluded.

This research was part of a master's dissertation. Due to logistical challenges, including a tight timeframe and limited logistical support, the review protocol was not pre-registered at that time. However, it was subsequently registered retrospectively on INPLASY. Due to these challenges, the search was conducted in four electronic databases and included only the published articles. The exclusion of abstracts, ongoing trials without published outcomes and grey literature might result in the exclusion of valuable insights from unpublished research and studies with negative findings [67], which might lead to an incomplete and potentially skewed overview of the literature. Studies included in this review may have been susceptible to publication bias.

The inclusion of only RCTs provided a robust foundation for evaluating the effectiveness of interventions, enhancing the internal validity of this research. However, the review

also highlighted a key limitation in the lack of sufficient evidence to draw definitive conclusions about the effectiveness of interventions. The scarcity of relevant RCTs conducted in LMICs limits the extent to which meaningful insights can be gleaned from the review's findings.

## Implications for research and practice

The current review emphasised the urgent need to design and implement more school-based preventive interventions for anxiety and depression, especially in LMICs where the need is the greatest. There is a rise in mental health problems, including anxiety and depression, in LMICs due to rapid urbanisation, environmental challenges and shifting social dynamics, making it critical to extend research beyond the HICs. In general, there is a need for cross culturally adapted intervention to prevent and treat depression and/ or anxiety among young people [68]. As mentioned before, the heterogeneity of outcome measures was highlighted as a significant issue in other systematic reviews that include RCTs on anxiety and depression in children and adolescents. This heterogeneity of outcome measures makes it difficult to compare findings across studies. Consequently, it becomes challenging to determine which interventions are effective for specific groups. Future trials should employ standardised measurement tools to enhance comparability and reliability. Additionally, longer-term follow-up period should be incorporated to allow for the assessment of intervention effectiveness in preventing anxiety and depression over time.

The absence of high-quality studies underscores the need for substantial enhancements in research rigor as well as the reporting of study methodologies and outcomes. Efforts should focus on implementing random sequence generation techniques and adequate allocation concealment, ensuring adequate blinding, providing sufficient information on missing data and conducting analyses that address data gaps. In order to minimise the impact of selective reporting bias, studies should endeavour to register their research protocols that include pre-specified plans for outcome analysis.

Future reviews should focus on addressing the limitations of this review, such as incorporating studies published in different languages and broadening the search scope to encompass relevant studies from diverse sources. Researchers could consider adopting more stringent inclusion criteria to enable a more robust assessment of the evidence.

Moreover, this review's alignment and disparities with prior literature highlighted the need to account for the contextual differences when implementing and evaluating interventions in LMICs. There exists a substantial gap in both the comprehensiveness and applicability of the evidence presented in the current review. It remains unclear how effective school-based prevention programs are in preventing anxiety and depression among children and adolescents in LMICs. Moving forward, it is imperative to expand upon existing school-based prevention programs for real-world implementation.

## Conclusion

The results indicate the potential effectiveness of school-based interventions in preventing depression in the children and adolescents in the LMICs. However, due to limited evidence and potential biases, it is not possible to draw conclusions about the overall effectiveness of these programs from the current review. Further research is necessary to gain a more comprehensive understanding of this critical topic. By leveraging the strengths of this review and addressing its limitations, researchers and policymakers can collaborate to enhance mental health outcomes for children and adolescents in LMICs.

## Supporting information

**S1 File. Search strategies for each database.**
(DOCX)

**S2 File. Risk of bias assessment.**
(DOCX)

**S3 File. PRISMA 2020 Checklist.**
(DOCX)

**S1 Table. Studies identified in search.**
(XLSX)

**S2 Table. Data extraction.**
(XLSX)

## Author contributions

**Conceptualization:** Sharone Zhameden Dieu Yin, Masuma Pervin Mishu.

**Data curation:** Sharone Zhameden Dieu Yin, Mei Ken Low, Masuma Pervin Mishu.

**Formal analysis:** Sharone Zhameden Dieu Yin, Masuma Pervin Mishu.

**Investigation:** Sharone Zhameden Dieu Yin, Masuma Pervin Mishu.

**Methodology:** Sharone Zhameden Dieu Yin, Masuma Pervin Mishu.

**Supervision:** Masuma Pervin Mishu.

**Visualization:** Sharone Zhameden Dieu Yin.

**Writing – original draft:** Sharone Zhameden Dieu Yin, Masuma Pervin Mishu.

**Writing – review & editing:** Sharone Zhameden Dieu Yin, Mei Ken Low, Masuma Pervin Mishu.

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
