## [Decision Letter · Decision Letter 0]

20 Sep 2024

PONE-D-24-20854School-based interventions to prevent anxiety and depression in children and adolescents in low- and middle-income countries: A systematic reviewPLOS ONE

Dear Dr. Mishu,

Thank you for submitting your manuscript to PLOS ONE. After careful consideration, we feel that it has merit but does not fully meet PLOS ONE’s publication criteria as it currently stands. Therefore, we invite you to submit a revised version of the manuscript that addresses the points raised during the review process.

This is a well-written manuscript with a very clear objetive. However, some minor changes are required.  Please submit your revised manuscript by Nov 04 2024 11:59PM. If you will need more time than this to complete your revisions, please reply to this message or contact the journal office at plosone@plos.org . Please include the following items when submitting your revised manuscript:

We look forward to receiving your revised manuscript.

Kind regards,

Patricia Moreno-Peral

Academic Editor

PLOS ONE

Journal Requirements: When submitting your revision, we need you to address these additional requirements. 1. Please ensure that your manuscript meets PLOS ONE's style requirements, including those for file naming. The PLOS ONE style templates can be found at https://journals.plos.org/plosone/s/file?id=wjVg/PLOSOne_formatting_sample_main_body.pdf and https://journals.plos.org/plosone/s/file?id=ba62/PLOSOne_formatting_sample_title_authors_affiliations.pdf 2. As required by our policy on Data Availability, please ensure your manuscript or supplementary information includes the following:  A numbered table of all studies identified in the literature search, including those that were excluded from the analyses.   For every excluded study, the table should list the reason(s) for exclusion.   If any of the included studies are unpublished, include a link (URL) to the primary source or detailed information about how the content can be accessed.  A table of all data extracted from the primary research sources for the systematic review and/or meta-analysis. The table must include the following information for each study:  Name of data extractors and date of data extraction  Confirmation that the study was eligible to be included in the review.   All data extracted from each study for the reported systematic review and/or meta-analysis that would be needed to replicate your analyses.  If data or supporting information were obtained from another source (e.g. correspondence with the author of the original research article), please provide the source of data and dates on which the data/information were obtained by your research group.  If applicable for your analysis, a table showing the completed risk of bias and quality/certainty assessments for each study or outcome.  Please ensure this is provided for each domain or parameter assessed. For example, if you used the Cochrane risk-of-bias tool for randomized trials, provide answers to each of the signalling questions for each study. If you used GRADE to assess certainty of evidence, provide judgements about each of the quality of evidence factor. This should be provided for each outcome.   An explanation of how missing data were handled.  This information can be included in the main text, supplementary information, or relevant data repository. Please note that providing these underlying data is a requirement for publication in this journal, and if these data are not provided your manuscript might be rejected. 3. Please review your reference list to ensure that it is complete and correct. If you have cited papers that have been retracted, please include the rationale for doing so in the manuscript text, or remove these references and replace them with relevant current references. Any changes to the reference list should be mentioned in the rebuttal letter that accompanies your revised manuscript. If you need to cite a retracted article, indicate the article’s retracted status in the References list and also include a citation and full reference for the retraction notice. 

Reviewers' comments:

Reviewer's Responses to Questions

**Comments to the Author**

1. Is the manuscript technically sound, and do the data support the conclusions?

Reviewer #1: Yes

Reviewer #2: Partly

2. Has the statistical analysis been performed appropriately and rigorously? 

Reviewer #1: N/A

Reviewer #2: N/A

3. Have the authors made all data underlying the findings in their manuscript fully available?

Reviewer #1: Yes

Reviewer #2: No

4. Is the manuscript presented in an intelligible fashion and written in standard English?

Reviewer #1: Yes

Reviewer #2: Yes

5. Review Comments to the Author

Reviewer #1: This is a well-written manuscript and the authors have a very clear objective which they carry out with the appropriate methodology for systematic reviews, including an assessment of the risk of bias and of the quality of the evidence. The results are discussed in a clear, organized manner and valuable conclusions are drawn. Considering that this is a systematic review with a narrative synthesis of the results no statistical analyses were performed. All data were available for review either in the main text or as supplementary documents.

Below are some additional comments for the authors or points for improvement in the text.

1. Method: Is there Prospero registration? If so, include the registration information and mention in the Method section.

2. In the narrative synthesis of the findings (270-370) study findings are discussed individually for each study. It would be helpful to also include a summary statement for each section (depression and anxiety, only depression, and only anxiety), that brings together the findings. For instance, in the narrative synthesis for depression it would be helpful to summarize in a sentence, either at the beginning or at the end, that both studies found that the interventions led to a reduction in depressive symptoms, although for one of them the reduction did not reach a level of significance.

3. The heterogeneity of outcome measures (450-451 and elsewhere in the text) especially in RCTs including minors has been noted in the literature before and highlighted as a problem in other systematic reviews—it would be helpful to highlight it further and cite some examples since it seems to be a big problem in being able to compare findings and ultimately address ‘what works for whom’ in terms of interventions.

4. The review focuses on school-based interventions in low and middle-income countries but the reviewed literature does not include any low-income countries since no studies were available for these. This finding deserves to be further highlighted in the discussion, not as a limitation of this review but as a worrisome finding for the lack of studies (and perhaps the lack of such interventions) in countries that may need it the most.

5. 488-491: It is unclear what is meant here by “the constraints imposed by the timeframe of this research” and how “given that studies with positive findings tend to have a higher likelihood of being published this review is susceptible to publication bias.” Perhaps rephrase. Is the intended statement that the studies included in this review may have been susceptible to publication bias?

And finally some typos in need of correction:

149: “condition” instead of “conditions”

155: “assessors” instead of “accessors”

163: only English language studies as a significant limitation since many of the LMICs (and perhaps local research) may not be in English

172: ideally include the precise date

194: “assessors” instead of “accessors”

276: “was observed” instead of “were observed”

323: “that involved” instead of “that involving”

368: “reduction” instead of “reductions”

374: “were due” instead of “was due”

394: “was deemed” instead of “were deemed”

405: remove “exist”

475: “reflects” instead of “reflects”

476: “interventions” instead of “intervention”

492: “the exclusion” instead of “the omission”?

504: remove “a” from “a longer-term”

516: remove “a” from “a more stringent inclusion criteria”

522: “school-based measures” or “school-based prevention programs”?

Reviewer #2: I agree that many, if not all, of the systematic reviews are done on studies from Western countries with sound economic income. Studies of anxiety and stress prevention in children and adolescents in countries outside these criteria are scarce, as this study has shown.

Suppose the reason they have chosen such studies in countries with middle or low-income levels is that it is assumed that there is a greater likelihood of pre-anxiety or pre-depressive symptoms simply because they have fewer resources. However, this assumption may be erroneous. The factors leading to these symptoms may differ between developed and developing countries. Social pressure is more significant in developed countries, and the protective role and values associated with the family may be higher in developing countries such as Brazil or China.

Some aspects should be taken into account:

1.- Table 2 points out the cut-off point of anxiety and/or depression that marks the measure used to understand the previous levels better.

2.- Does this review require a code of Prospero or similar? This condition is usually necessary before its acceptance.

3.- To justify the inclusion of these countries, ILMCs should have gone more profound in the discussion section of the similarities and differences with other reviews that would have included developed Western countries. They may share explanatory mechanisms, and differences in different aspects may also be observed.

Otherwise, the work meets the standards of a systematic review and is pleasant and comprehensive to read.

6. PLOS authors have the option to publish the peer review history of their article (what does this mean? ). If published, this will include your full peer review and any attached files.

**Do you want your identity to be public for this peer review?** For information about this choice, including consent withdrawal, please see our Privacy Policy .

Reviewer #1: No

Reviewer #2: **Yes: ** Jose M Mestre

---

## [Author Response · Author response to Decision Letter 1]

4 Dec 2024

Response to Reviewers comments:

We would like to thank the reviewers for their helpful comments. We addressed all the comments and incorporated the suggestions. Please see the attached Document 'Response to reviewers) that includes the responds to each point raised by the academic editor and reviewer(s).

All line references are based on the document “Revised Manuscript with Tracked Changes”.

---

## [Editor Report · Decision Letter 1]

18 Dec 2024

School-based interventions to prevent anxiety and depression in children and adolescents in low- and middle-income countries: A systematic review

PONE-D-24-20854R1

Dear Dr. Masuma Pervin Mishu, 

We’re pleased to inform you that your manuscript has been judged scientifically suitable for publication and will be formally accepted for publication once it meets all outstanding technical requirements.

Kind regards,

Patricia Moreno-Peral

Academic Editor

PLOS ONE

---

## [Editor Report · Acceptance letter]

PONE-D-24-20854R1

PLOS ONE

Dear Dr. Mishu,

I'm pleased to inform you that your manuscript has been deemed suitable for publication in PLOS ONE. Congratulations! Your manuscript is now being handed over to our production team.

Kind regards,

on behalf of

Dr. Patricia Moreno-Peral

Academic Editor

PLOS ONE